# Recent Progress of Magnetic Nanomaterials with Enhanced Enzymatic Activities in Antitumor Therapy

**DOI:** 10.3390/ijms262210890

**Published:** 2025-11-10

**Authors:** Yifan Zhang, Dongyan Li, Hongxia Liang, Bin Lan, Peidan Chang, Yaoxin Yang, Yuanyuan Cheng, Galong Li, Hongbing Lu

**Affiliations:** 1College of Life Sciences, Northwest University, Xi’an 710069, China; yifanzhang@sntcm.edu.cn (Y.Z.);; 2Institute for Chinese Medicine Frontier Interdisciplinary Science and Technology, Shaanxi University of Chinese Medicine, Xianyang 712046, China; 3School of Biomedical Engineering, Air Force Medical University, Xi’an 710032, China

**Keywords:** MNPs, nanozymes, immobilized enzymes, catalytic activity, anticancer therapy

## Abstract

Magnetic nanomaterials with enhanced enzymatic activities have garnered significant attention from researchers worldwide. Magnetic nanomaterials, including nanozymes and immobilized enzymes, can initiate specific catalytic reactions in the diseased microenvironment for cancer treatment. In this review, we aim to present the significant advancements in synthesizing various types of magnetic nanomaterials with enhanced enzymatic activities and their antitumor therapy applications in the past five years. We first show the representative magnetic nanomaterials and elucidate their fundamental mechanisms related to magnetic properties and electromagnetic effects (such as magneto-thermal, magneto-mechanical, and magneto-electric effects). Secondly, we introduce magnetic nanozymes and magnetic immobilized enzymes and discuss the creative methods allowing the enzymatic activities of nanomaterials to be remotely enhanced by various electromagnetic effects. We also discuss some innovative magnetic nanomaterials that exhibit unique responsiveness to external energies (such as X-rays and ultrasounds) for killing cancer cells. Finally, we address future research suggestions in rationally designing advanced magnetic nanomaterials with remote increased enzymatic activities and discuss challenges and opportunities for efficient cancer therapy.

## 1. Introduction

The global cancer burden continues to rise. There were approximately 18.5 million new cancer diagnoses and 10.4 million cancer deaths in 2023, resulting in a total of 271 million disability-adjusted life-years (DALYs). The number of cancer diagnoses and deaths is projected to rise substantially from 2024 to 2050, by 60.7% and 74.5%, respectively. This will result in an estimated 30.5 million new cancer diagnoses and 18.6 million deaths by 2050 [1,2].

Magnetic nanomaterials with enzymatic activities, especially magnetic nanozymes and immobilized enzymes, are bringing about a new revolution in cancer catalytic therapy [3,4,5,6]. This therapeutic approach employs magnetic nanomaterials to trigger specific catalytic reactions in pathological regions, thereby modulating metabolic levels in vivo and achieving the goals of cancer treatment [7,8]. Specifically, these magnetic nanomaterials, owing to their inorganic crystal structure, easy reusability, and low cost for scale-up, demonstrate the advantages of high efficiency, selectivity, and stability. Compared to natural enzymes, magnetic nanozymes and immobilized enzymes can overcome the limitations of natural enzymes, such as restricted operational conditions and poor stability [9]. Thus, magnetic nanomaterials are key components. Progress in nanoscience and nanotechnology has enabled the improvement of magneto-responsive nanomaterials by changing their composition, size, morphology, and surface modification [3,10]. Various magnetic nanomaterials have been designed and synthesized in the past five years, including different compositions (such as Fe_3_O_4_, MnFe_2_O_4_, CoFe_2_O_4_, ZnFe_2_O_4_, and CoFe_2_O_4_@MnFe_2_O_4_) and morphologies (such as nanoparticles, nanocubes, nanorings, and nanoflowers) [5,7,11].

Significant progress has been made in enhancing the enzymatic activities of magnetic nanomaterials, which has greatly contributed to efficient cancer catalytic therapy [6]. As magnetic fields can pass through the tissues with no tissue penetration depth limitations and no radiation hazards, magnetic nanomaterials can act as nanoscale transducers of magnetic fields to generate electromagnetic effects, including magneto-thermal, -mechanical, and -electric effects [12,13]. These effects show high temporal and spatial resolution and controllability in arbitrarily deep diseased tumor regions [14]. Other external physical energies (such as X-ray and ultrasound) have also been applied to increase the enzymatic reaction of magnetic nanomaterials, which has great potential in the treatment of deep-seated tumors. Magnetic nanomaterials hold promise for replacing natural enzymes in the field of precision catalytic medicine. Therefore, a comprehensive understanding of the design of magnetic nanomaterials with remotely increased enzymatic activities in antitumor therapy is required.

In this review, we aim to present recent significant advancements in various versatile magnetic nanomaterials with enhanced enzymatic activities for efficient anticancer applications (Figure 1). We first elucidate the fundamentals related to magnetic properties and electromagnetic effects of the nanomaterials. Secondly, we introduce the design strategies and catalytic mechanisms of the nanomaterials, including nanozymes and immobilized enzymes. We review the particular advancements in magnetic nanomaterials with increased enzymatic activities under various magnetic fields for antitumor therapy in the past five years. We also discuss some innovative magnetic nanomaterials that exhibit unique responsiveness to external energies (such as X-ray and ultrasound). In addition, the challenges of the uncertainties in the existing research are discussed. Finally, we address future research suggestions in rationally designing advanced magnetic nanomaterials with remote increased enzymatic activities and discuss challenges and opportunities for efficient cancer therapy.

## 2. Various Magnetic Nanoparticles (MNPs) and Their Electromagnetic Effects

MNPs with dimensions at the nanometer scale (1–100 nm) exhibit unique magnetic field-responsive properties. In the biomedical field, the most widely used MNPs are primarily categorized into three groups, including metals (e.g., Co, Fe, Mn, Ni), metal oxides (e.g., Fe_3_O_4_, Mn_3_O_4_, MnFe_2_O_4_), and magnetic nanocomposites (e.g., Fe_3_O_4_@GOx, FePt-FeC, CoFe_2_O_4_-BiFeO_3_). Among these, magnetic Fe_3_O_4_ nanoparticles with excellent biocompatibility have been extensively studied for various biomedical applications, including tumor hyperthermia, magnetic resonance imaging, drug delivery, and neurostimulation [15]. For example, superparamagnetic Fe_3_O_4_ nanoparticles are injected for the neoadjuvant chemotherapy of Osteosarcoma (NCT04316091) [8] and magnetic beads loaded with doxorubicin are delivered for the magnetic-guided chemotherapy of colorectal neoplasm (NCT00041808).

The magnetic properties of MNPs that can be finely tuned are pivotal to their biomedical applications. Key parameters for characterizing these properties include saturation magnetization (Ms), magnetic anisotropy (K), remanence (Mr), and coercive force (Hc). The dimensions and morphology of magnetic nanomaterials have a great impact on their electromagnetic properties. Optimizing the dimensions and morphology of nanomaterials is crucial to improving their electromagnetic properties and biomedical applications. Chemical synthesis methods such as thermal decomposition and hydrothermal reaction are commonly used [3,7]. The size of nanoparticles is closely associated with the reaction time and the boiling point of the organic solvent during thermal decomposition. Controlling reaction time and temperature can restrict the overgrowth of nanoparticles. Using organic ligands (such as oleylamine) can shape their morphology.

Apart from superior magnetic properties of MNPs, efficient antitumor therapy underscores the need for imaging the sufficient accumulation of MNPs within tumors. Magnetic resonance imaging (MRI) enables the high-contrast imaging of tumor tissue. The whole-body MRI is a promising noninvasive modality for cancer diagnosis [16,17]. However, WB-MRI still possesses modest detection rates. MNPs, one of the most used nanoscale imaging tools, can serve as MRI contrast agents for improving cancer detection rates. According to biomarkers of cancer, the MNPs can be modified with various antibodies or molecules and then actively targeted to tumors following administration. MRI can be used to confirm the maximum accumulation of the nanoparticles in the tumor site, which is an important step for improving the efficacy of nanoparticle-catalyzed antitumor therapy.

Magnetic properties allow MNPs within the tumor to respond to specific magnetic fields, producing distinct electromagnetic effects such as magneto-thermal effects, magnetic force effects, and magneto-electric effects for enhancing enzymatic activity (Figure 2). The used magnetic field with magnitudes in the millitesla range can pass through tissue without attenuation and harmful effects, relying on the negligibly low conductivity and magnetic susceptibility of biological tissue. Therefore, the magnetic field-induced electromagnetic effects of nanoparticles represent an extremely ingenious approach to achieve efficient noninvasive cancer therapy.

### 2.1. Magneto-Thermal Effect of MNPs

The magneto-thermal effect describes the process by which MNPs generate local heat under an alternating magnetic field (AMF) with a frequency ranging from 100 kHz to 1 MHz. This energy conversion leads to localized temperature elevation in the vicinity of the nanoparticles. For iron-based particles with diameters in the tens of nanometers, heat production results from the synergistic effects of hysteretic loss as well as Néel and Brownian relaxation. In contrast, superparaMNPs with smaller diameters predominantly rely on Néel relaxation for thermal energy conversion. The magneto-thermal efficiency of MNPs is generally evaluated by specific absorption rate (SAR) (or specific loss power), which is defined as the rate of heating of a certain mass of MNPs within a limited period. The unit of SAR is W/g. The higher the values of SAR are, the stronger the magnetic heat efficiency will be. The magneto-thermal efficiency of nanoparticles is governed by a dual set of factors, including intrinsic particle properties (e.g., size, morphology, composition, and surface modifications) and external magnetic field parameters (e.g., frequency, amplitude, and duration). This interdependence highlights the need for optimized nanoparticle design and magnetic field parameter tuning to achieve efficient heating performance. The Ms and SAR of superparamagnetic nanoparticles exhibit a size-dependent relationship. Larger particle sizes generally lead to higher Ms values and enhanced SAR values. For a given MNP system, the SAR value is proportional to the square of the magnetic field intensity (H) and the magnetic frequency (*f*). This dual dependence underscores the importance of nanoparticle size optimization and field parameter tuning for maximizing thermal efficiency in biomedical applications. Modifying the physical and chemical properties of MNPs to enhance their magnetic characteristics, such as Ms, anisotropy, and thermal conductivity, is a fundamental strategy to improve the SAR and SLP.

### 2.2. Magneto-Mechanical Effect of MNPs

The magneto-mechanical effect refers to the mechanical effect produced by nanoparticles when they are subjected to low-frequency (<100 Hz) static magnetic fields, gradient magnetic fields, or rotating magnetic fields [18]. This effect is caused by the interaction between the magnetic field gradient and the magnetic moment of the particles [19]. The magnitude and direction of magneto-mechanical effect (force and torque) are jointly determined by the characteristics of MNPs (such as size, morphology, composition, and saturation magnetization) and the parameters of the exposed magnetic field (such as intensity, gradient, and direction) [20].

### 2.3. Magneto-Electric Effect of MNPs

The magneto-electric effect refers to an energy conversion phenomenon between magnetism and electricity, where MNPs exhibit electric charge polarization under an AMF [21]. Based on different generation mechanisms, the magneto-electric effect can be classified into the direct magneto-electric effect and the piezomagneto-electric effect [22]. The direct magneto-electric effect is mainly mediated by single-phase magneto-electric materials (such as BiFeO_3_ and BaMnF_4_) to generate an external magnetic field [21]. Due to its simultaneous presence of ferroelectricity and ferromagnetism, it can exhibit weak magneto-electric properties at low temperatures. The piezoelectric-electromagnetic effect refers to the magnetic-electrical composite material formed by combining magnetostrictive materials and piezoelectric materials in a certain way [23]. Under an external magnetic field, it generates the magneto-electrical effect through the coupling effect of the magnetostrictive effect and the piezoelectric effect. The magnetoelectric properties of magneto-electric composite materials are usually measured by the magnetoelectric voltage coefficient, which indicates the degree of the “electric field–magnetic field” coupling in the material [22]. The larger the value, the higher the efficiency of energy conversion between magnetism and electricity in the material. The magnetostrictive-piezoelectric composite material can generate a much higher magnetoelectric effect than that of a single-phase magnetoelectric material due to the synergistic effect.

## 3. Design and Synthesis of Magnetic Nanomaterials with Enzymatic Activities

### 3.1. Magnetic Nanozymes

The term “nanozymes” as “nanomaterials with enzyme-like characteristics” was further formally defined in 2013 and established the conceptual foundation of this field. Among the various nanozymes, magnetic metal oxide nanozymes have garnered much research interest. These nanozymes are typically constituted of a magnetic metal oxide core and a surface modification shell. Precise control over their dimensions (2–100 nm) and geometries (e.g., spheres, cubes, nanorods, nanorings, or two-dimensional nanosheets) can be achieved by tailoring straightforward synthesis conditions, such as employing thermal decomposition, hydrothermal methods, and coprecipitation methods. The subsequent surface modification of nanoparticles with additives can further enhance nanozymes’ stability, biocompatibility, water dispersibility, targeting ability, and oxidation resistance [24]. Thus, magnetic metal oxide nanozymes are emerging as an important subclass that effectively overcomes the limitations of natural enzymes. These nanozymes, compared to enzymes, have demonstrated their superior catalytic performance, including high stability, easy reusability, and low cost. Furthermore, the catalytic performance of the nanozymes can be significantly enhanced by electromagnetic effects under various magnetic fields. In the past five years, various magnetic nanozymes have been created, aiming to replicate the structural and functional complexity of natural enzymes, particularly in order to improve the nanozymes’ catalytic efficiency under complex physiological conditions such as the tumor microenvironment.

While single-component magnetic nanozymes benefit from structural simplicity and straightforward preparation, they also suffer from several limitations, including restricted catalytic functionality and a tendency toward surface oxidation. In particular, the nanoparticles are prone to agglomeration, leading to a reduction in active sites and degradation of the catalytic performance. Moreover, in oxidative reaction environments, the surfaces of nanoparticles are easily oxidized, forming a passivation layer that compromises catalytic efficiency and selectivity. These drawbacks have motivated the development of more sophisticated multicomponent nanocatalyst systems.

Multicomponent magnetic nanozymes have been developed to overcome the above-mentioned drawbacks. For example, core–shell structured magnetic nanozymes, composed of an MNP core and a surface modification shell, are designed not only to protect the core from chemical corrosion and oxidation but also to provide a tunable surface shell and abundant active sites. By judicious selection of shell materials and precise control over shell thickness, the catalytic performance, targeting, and stability properties of these nanomaterials can be finely tuned. Silica (SiO_2_) is one of the most widely employed coating materials for core–shell structured nanocatalysts, valued for its exceptional chemical inertness, versatile surface modifiability, and controllable porous morphology [25,26]. By forming a dense and uniform layer around the magnetic core, SiO_2_ effectively shields the MNPs from dipole–dipole interactions, thereby suppressing aggregation and oxidation of the core material. Furthermore, the abundant surface silanol groups facilitate the construction of multifunctional hybrid nanocomposites. Studies indicate that thinner layers (ca. 5–10 nm) promote reactant diffusion but provide limited protection to the core, whereas thicker coatings (ca. 20–30 nm) significantly enhance stability but may introduce mass transfer limitations. In addition, the inherent hydrophilicity of silica supports the formation of stable colloidal systems and enables facile surface functionalization. The porous silica architecture also contributes to high specific surface area and reduced optical reflectivity, extending its utility across various catalytic and photochemical applications. Another study also demonstrated that carbon-encapsulated nickel ferrite (NiFe_2_O_4_@C) nanocatalysts can efficiently decompose hydrogen peroxide (H_2_O_2_) within the tumor area to generate hydroxyl radicals (•OH), achieving effective tumor-killing effects.

Multicomponent magnetic nanozyme frameworks that incorporate nanoparticles into covalent organic frameworks (COFs) or metal organic frameworks (MOFs) leverage the synergistic effects between the components [27,28]. This kind of magnetic nanocatalyst exhibits superior performance compared to its individual counterparts. It also displays a large specific surface area, tunable pore structures and functionality, and abundant catalytically active sites. The framework matrix can prevent the aggregation, oxidation, or corrosion of MNPs, thereby extending their operational lifespan [29]. Such MOF/COF-based magnetic nanocatalysts demonstrate broad application potential in cancer catalytic therapy, drug delivery, photocatalysis, electrocatalysis, and heterogeneous organic synthesis.

#### 3.1.1. Fe_3_O_4_ Nanozymes with Peroxidase-like Activity

Magnetite (Fe_3_O_4_) nanoparticles, as a prototypical magnetic metal oxide nanozyme, were unexpectedly discovered to exhibit peroxidase-like activity in 2007, which marked the emergence of MNPs as enzyme mimics [30]. Fe_3_O_4_ nanoparticles have a unique cubic inverse spinel crystal structure. Fe^2+^ cations occupy octahedral sites, while Fe^3+^ cations are uniformly distributed between octahedral and tetrahedral coordination sites. O^2−^ anions fill the interstitial lattice positions, forming a highly ordered framework. This distinctive atomic arrangement not only endows Fe_3_O_4_ with unique magnetic properties but also offers excellent catalytic activity. Fe_3_O_4_ nanozymes with superior chemical stability address the bottleneck issue of stability and recycling encountered with natural enzymes, thereby showing great potential in biomedical applications.

The catalytic mechanisms of Fe_3_O_4_ nanozymes are some of the most extensively studied mechanisms. Studies have shown that Fe_3_O_4_ nanozymes under acidic pH (3–6.5) exhibit catalytic behavior analogous to that of horseradish peroxidase (HRP). Fe_3_O_4_ nanozymes can catalyze the oxidation of substrates (such as 3,3′,5,5′-tetramethylbenzidine), resulting in a colorimetric response similar to that mediated by natural peroxidases [31]. At the same time, Fe_3_O_4_ catalyzes H_2_O_2_ to form free radicals (predominantly •OH) and oxidize hydrogen donors to generate H_2_O. In particular, Fe_3_O_4_ nanozymes were found to mimic the Michaelis–Menten kinetics and ping-pong catalytic characteristics of natural enzymes [32]. Since then, various nanostructures have been developed as HRP mimics, including metal-based materials, conductive polymers, carbon nanomaterials, and single-atom catalysts. In particular, the HRP-like catalytic mechanism of Fe_3_O_4_ was divided into four steps: a surface Fenton-like reaction, internal electron transfer, the outward migration of excess Fe ions, and chemical composition changes (Figure 3) [33].

However, subsequent studies indicated that in addition to free •OH, surface-bound •OH and iron-oxo species with high valent (e.g., Fe=O) also play crucial roles in catalysis. Maxim et al. demonstrated through spin trapping–EPR comparative experiments that radical generation primarily occurs via surface-mediated catalysis rather than via dissolved metal ions, as previously assumed. Their data further revealed that surface catalytic sites on nanoparticles are at least 50 times more efficient in generating •OH than dissolved Fe^3+^ ions. Wan et al. reported that due to the reduced state of surface iron atoms on Fe_3_O_4_ nanoparticles, the energy barrier for O–O bond cleavage in H_2_O_2_/•OOH is low (0.14 eV). Combined with the highly oxidizing surface-formed Fe=O species (0 eV barrier), both the •OH pathway and the Fe=O pathway are thermodynamically favored. High-valent iron-oxo species are also key intermediates in the catalytic reaction of peroxidases (PODs). It is noteworthy that bioinspired synthesis is an extremely effective method to increase the catalytic performance of nanozymes. For instance, the active sites of nature HRP exhibit structural diversity, involving elements such as heme, cysteine residues, selenium, and manganese. The Anteneh group fabricated the Fe_3_O_4_–Fe_0_/Fe_3_C nanozyme via carbothermal reduction, which not only optimized the Fe^2+^/Fe^3+^ cycle and electron transfer to promote •OH generation but also enhanced TMB adsorption through the Lewis acidity of Fe^3+^.

#### 3.1.2. Magnetic Nanozymes with Catalase-like Activity

Magnetic nanozymes with catalase (CAT)-like activity have attracted significant research interest due to their ability to catalyze the transformation from H_2_O_2_ into O_2_ and H_2_O. This protects cells from hypoxia and oxidative stress. Fe_3_O_4_ nanozymes show CAT-like activity under neutral and basic pH (7–10), except for their HRP-like activity. For example, Zhang et al. fabricated a magnetic iron oxide nanocomposite via a biomineralization method. The nanozymes exhibited CAT-like activity and a high SAR of 2390 W/g [34].

Elucidating the underlying catalytic mechanisms is crucial for advancing CAT-like magnetic nanozymes. A series of theoretical studies were conducted to unravel the catalytic pathways according to density functional theory. In general, H_2_O_2_ contains H–O and O–O chemical bonds. The CAT-like catalytic mechanisms of nanozymes can be categorized into two types: heterolytic and homolytic cleavage. The heterolytic cleavage mechanism suggests that H_2_O_2_ molecules preferentially undergo H–O bond breakage, yielding protons and hydroxyl adsorbates. In contrast, the homolytic cleavage mechanism proposes that H_2_O_2_ first undergoes O–O bond dissociation, decomposing uniformly into two hydroxyl adsorbates. Guo et al. studied the potential catalytic mechanisms of such CAT-like activity at the atomic or molecular level, including alkaline dissociation, acidic dissociation, and the double-peroxide association pathway. Based on calculations of thermochemical energies and corresponding activation energy barriers, they proposed that the double-peroxide association mechanism is the most feasible pathway for the CAT-like catalytic cycle of Co_3_O_4_.

#### 3.1.3. Magnetic Nanozymes with Oxidase-like Activity

A variety of metal oxide-based magnetic nanozymes capable of mimicking the function of natural oxidases (OXDs) have been reported in recent years. They facilitate substrate oxidation by promoting electron transfer to O_2_, accompanied by the generation of other oxidized products and reactive oxygen species (ROS). Depending on the functional groups of the donor substrates, these nanozymes can be categorized into types acting on amino groups, CH-OH groups (e.g., glucose oxidase, GOx), phenolic hydroxyl groups, among others. Among these, oxidase nanozymes targeting amino groups have been the most extensively studied.

Significant progress has been made in understanding the OXD-like catalytic mechanisms of magnetic nanozymes. Studies indicate that reaction intermediates (e.g., singlet oxygen, superoxide anions) and electron transfer pathways play critical roles in their OXD-like properties. For instance, Zhang et al. reported that Mn_3_O_4_ nanoparticles mediate electron transfer to O_2_ via surface Mn^2+^/Mn^3^ redox cycling, leading to the generation of superoxide anions (•O_2_^−^). These intermediates can be partially converted to H_2_O_2_ through either non-enzymatic disproportionation or superoxide dismutase-catalyzed reactions, with subsequent decomposition assisted by transition metal ions to yield •OH. Subsequently, •O_2_^−^, •OH, and high-valent manganese species collectively participate in substrate oxidation, often accompanied by a colorimetric response. Wang et al. systematically studied the OXD-like activity of Mn_3_O_4_ nanoparticles, demonstrating that they facilitate the oxidation of TMB to form bluish-green oxTMB via ROS. Introducing arsenic onto the material surface was shown to reduce high-valent manganese species, significantly enhancing the OXD-mimicking activity.

#### 3.1.4. Magnetic Nanozymes with Multiple Enzymatic Activities

Magnetic nanozymes, especially tmulticomponent magnetic nanozymes, often combine multiple catalytic activities. This might be due to their relatively low specificity, unlike enzymes. These nanozymes exhibit simultaneous pro-oxidant and antioxidant activities (Figure 4). The multiple catalytic activities of magnetic nanozyme composite are often functionally significant for improving anticancer therapeutic efficiency. A significant proportion of reported magnetic nanozymes focus exclusively on POD activity, often without evaluating closely related enzymatic activities (such as OXD and CAT). Panferov et al. found that many magnetic nanozymes, like metal oxide nanozymes, show simultaneous POD, OXD, and CAT activities [32]. As confirmed, the OXD activity of Mn_2_O_3_ nanozymes exceeds the POD activity, explaining the previously observed phenomena of decreased catalytic activity for Mn oxide nanozymes. Therefore, it is important to comprehensively characterize the enzymatic profiles of magnetic nanozymes rather than focusing on single-activity findings.

### 3.2. Magnetic Nanomaterial-Immobilized Enzymes

Natural enzymes have many distinct advantages, including superior catalytic activity, high substrate specificity, and selectivity under biological mild reaction conditions. Unfortunately, the free enzymes within an unfavorable microenvironment (e.g., temperature, pH, mass transfer resistance) show decreased stability and catalytic activity, resulting in high costs and low production efficiency. These drawbacks limit their anticancer efficiency.

MNPs have unique advantages for immobilizing and utilizing enzymes, such as a large surface area, simple surface modification, water dispersibility, and easy separation using external magnetic fields. In previous work, various MNPs with different chemical compositions and morphologies have been employed to immobilize enzymes such as lipase, horseradish peroxidase, glucose oxidase, lactate dehydrogenase, dehalogenase L-2-HADST, chymotrypsin, and β-galactosidase. The commonly used methods for immobilizing enzymes include covalent binding, non-covalent adsorption, and encapsulation. The MNP-based immobilization strategy endows enzymes with improved operational stability, reusability, and recyclability for long-term anticancer catalytic therapy. For example, after immobilization on magnetic nanoparticles, the enzymatic chimera that was formed by D-amino acid oxidase (DAAO) and N-acetylmuramoyl-L-alanine amidase (CLytA) offered some advantages compared to the free CLytA-DAAO, including increased stability, extended action time, and enhanced cytotoxicity to pancreatic and colorectal carcinoma as well as glioblastoma [35].

Based on unique electromagnetic effects (e.g., magneto-thermal, -mechanical, and -electrical effects), MNPs have demonstrated their pivotal roles in the remotely regulated and enhanced catalytic activities of immobilized enzymes. This is because enzymatic activities have a universal dependency on temperature, protein conformation, and electron transfer. Several important studies have confirmed that the magneto-thermal effect of magnetic nanomaterials can significantly increase enzyme activity. An important mechanism was recently reported: thermal photon currents contribute to the magneto-thermal activation of the enzyme under radiofrequency magnetic fields [36]. More detailed discussions about the strategies for enhancing the activities of immobilized enzymes for improving anticancer applications are presented in Table 1 and Section 4.

## 4. Enhanced Enzymatic Activities of Magnetic Nanomaterials for Efficient Antitumor Therapy

### 4.1. Enhanced Catalytic Activity of Magnetic Nanomaterials by Magneto-Thermal Effect

The rate of enzymatic activity is significantly affected by the reaction temperature. Within a certain range, the reaction rate increases as the temperature rises. It is generally believed that increasing the temperature enhances the reaction rate by increasing the probability of molecular collisions and the proportion of activated molecules. Therefore, many MNPs capable of generating a magneto-thermal effect are further used to enhance the catalytic reaction rate of nanoparticle-based nanozymes and nanoparticle-immobilized enzymes by increasing the surface temperature of the nanoparticle or by raising the solution temperature [38].

He et al. investigated changes in the HRP-like activity of Fe_3_O_4_ nanozymes with different diameters and morphologies under an AMF [39]. The experiment revealed that the higher the SAR value of the Fe_3_O_4_ nanozymes, the greater the amplitude of the catalytic activity. This work revealed that the SAR of the nanoparticles is the key factor influencing the performance of magnetic nanozymes. Further, the same group reported a new type of vortex magnetic domain structure, called vortex magnetic iron oxide (FVIO). The SAR value of the vortex Fe_3_O_4_ nanozyme is one order of magnitude larger than that of superparamagnetic Fe_3_O_4_ nanoparticles under AMF (at 400 kHz and 740 Oe) [48]. FVIO modified with graphene oxide (GO) nanosheets showed a significantly higher SAR value of 5054 W/g [40]. Thus, FVIO nanozymes can remarkably enhance the production of ROS in mouse tumor tissues, improving the catalytic therapy efficiency for 4T1 breast cancer. Notably, Shen et al. further revealed the mechanism of magnetic heat activation of nanoenzyme activity based on an iridium-functionalized manganese ferrite nanoparticle, the Ir@MnFe_2_O_4_ nanoparticle [41]. The iridium (III) complex (Ir) provides mitochondrial targeting for the nanoparticles, while the MnFe_2_O_4_ nanozyme has both peroxidase catalytic activity and magnetic heat effect. Under an AMF, the Ir@MnFe_2_O_4_ nanozymes increased local temperature, which accelerated the reduction of surface Fe(III) on nanoparticles to Fe(II) by glutathione (GSH) and promoted the generation of •OH from H_2_O_2_, leading to more efficient cancer treatment.

The catalytic activity of nanoparticle-immobilized enzymes can also be improved using the magnetic heating effect. For natural enzymes, within a certain range, increasing the temperature can also accelerate the reaction rate by increasing the thermal energy of the substrate/enzyme molecules and enhancing molecular collisions. However, higher temperatures can also disrupt many of the interactions that maintain the three-dimensional structure of enzymes (such as van der Waals forces, hydrogen bonds, and ionic bonds), thereby reducing the enzyme’s activity. Previous studies have presented many immobilized enzymes with MNPs as carriers through methods such as electrostatic adsorption, physical embedding, and covalent immobilization. According to some early works, the magnetic thermal regulation of enzyme activity mainly depends on the macroscopic temperature changes in the reaction solution where the enzyme molecules are located. Due to the weak magneto-thermal conversion (SAR of approximately 250 W/g) of superparamagnetic Fe_3_O_4_ nanoparticles, the effectiveness of the magnetic thermal regulation of enzyme activity still needs to be optimized. To enhance the regulatory effect, researchers often adopt strategies such as increasing the dosage of magnetic particles or modifying the magnetic field parameters (such as extending the field exposure time or increasing the field strength). However, if the dosage or magnetic field parameters exceed a certain range, toxic side effects will inevitably occur in normal tissues.

Developing MNPs with enhanced SAR value is the key to improving the accuracy and safety of enzyme-mediated therapy. For example, Fe_3_O_4_ NR-immobilized glucose oxidase (Fe_3_O_4_ NRs@GOx), a cascade nanozyme-enzyme multicomponent catalyst, was constructed using polyethylene glycol as a bridge molecule with different molecular weights (different connection distances), as shown in Figure 5 [42]. The magnetic thermal effect of Fe_3_O_4_ NRs can simultaneously increase the activities of the POD nanozyme and GOx, thereby influencing the cascade reaction rate of the nanocatalysts. Under an AMF, the kinetics match of POD-GOx cascade nanocatalysts at an optimal polyethylene glycol length of 1 nm is the best, and the cascade activity is more than 400 times higher than that of the free GOx and POD nanozyme mixture. Both in vitro and in vivo experiments demonstrated a more significant ROS production and therapeutic effect on tumors. This work is the first to demonstrate that by ingeniously designing distance-dependent magnetic nanozyme-enzyme cascade catalysts, it is possible to precisely regulate the kinetics match of cascade catalytic reactions for efficient anticancer therapy. Therefore, magnetic regulation is a smart way to achieve the kinetic match of the cascade reaction [25,43].

Apart from ROS, the magnetic regulation of enzymes has also been applied to the activation of the original drug for achieving catalytic treatment for tumors. Torres-Herrero et al. employed the biological silicification method to co-package MNPs and HRP within biomimetic silica [44]. The HRP can convert the original drug molecule indole-3-acetic acid (3IAA) into peroxide-free radicals, triggering the oxidative degradation of DNA and proteins within the tumor, thereby promoting cell apoptosis. Under an AMF, the magnetic heating effect of nanoparticles accelerates the generation of peroxide free radicals. As a result, the cell death rate after AMF treatment reaches as high as 90%, while the cell death rate in the AMF-untreated group is only 10%. The group with AMF treatment also demonstrated superior therapeutic effects on pancreatic cancer tumors in mice. Therefore, magnetic heat-controlled prodrugs of enzymes hold great potential in enhancing anticancer therapeutic effects while reducing the toxic side effects of the drugs.

### 4.2. Enhanced Catalytic Activity of Magnetic Nanomaterials by Magneto-Mechanical Effect

SuperparaMNPs undergo rotational and vibrational movements in a low-frequency magnetic field, which can generate tensile, torsional, and bending forces. When these forces are transmitted to the enzyme molecules attached to the nanoparticles, it will cause changes in their secondary structure, thereby affecting the catalytic activity of the enzyme. Veselov et al. covalently coupled *α*-chymotrypsin (CT) to Fe_3_O_4_ nanoparticles (with a diameter of 25 nm) and thus synthesized Fe_3_O_4_-CT connected by amino groups [49]. By controlling the reaction conditions, each enzyme molecule can be simultaneously coupled with multiple MNPs. After the Fe_3_O_4_-CT nanocatalysts were exposed to a non-heated low-frequency magnetic field (*f* = 50 Hz, B = 140 mT, on 1 min, off 30 s, repeated 3 times), the apparent *K*_cat_ did not show significant changes, but the *K*_m_ value increased approximately two times. The content of the *α*-helix of the immobilized enzyme significantly decreased after the magnetic field stimulation, as demonstrated by the attenuated total reflection Fourier transform infrared spectroscopy. The molecular dynamics results also indicated that the substrate binding pocket of the enzyme molecules became smaller after the magnetic field stimulation. It is worth noting that 3 h after the removal of the magnetic field, the spatial conformation of the Fe_3_O_4_-immobilized enzyme was completely restored. This indicates that the magnetic force of Fe_3_O_4_ nanoparticles under magnetic fields is transmitted to enzyme molecules and affects the enzyme’s conformation. Besides enzyme molecules, DNAzymes can also be regulated by magnetic force [50]. As demonstrated in a “glass slide-DNAzyme-nanoparticle” system [51], the DNAzyme retains its catalytically active stem-loop structure without the application of a magnetic field. However, when a magnetic field is applied from different directions, the magnetic force causes the DNAzyme to stretch, thereby losing its spatial structure and leading to the inactivation of the DNAzyme. The magnetic nanoscale particles were able to achieve multiple and reversible switching regulation of the catalytic activity of the DNAzyme by an external magnetic field.

The regulating effect of enzymatic activity can also be achieved through magnetically changing the interaction between the substrate and the enzyme. This is because the interaction and collision between enzyme molecules and substrate molecules are prerequisites for a catalytic reaction to occur. When a low-frequency AMF is applied, the magnetic moments of the superparaMNPs tend to align in the direction of the field, thereby experiencing an action force in that direction [12]. Under the influence of low-frequency AMF, the nanoparticles will attempt to align along the constantly changing magnetic field direction, thereby achieving microscopic stirring behavior where the movement direction of the nanoparticles keeps changing. This “microscopic stirring” action enhances the efficiency of nanoparticle, enzyme molecule, and substrate collisions and diffusion in the solution, resulting in an increased reaction rate of the catalytic process. The magnetic effect can also regulate the catalytic activity of enzyme molecules by altering the microenvironment near the substrate-binding pocket of the enzyme, further influencing the enzyme–substrate interaction. Szekeres et al. covalently immobilized amyloglucosidase (AMG), urease (Ur), and esterase (Est) on superparaMNPs, with the PAA-b-PEGMA polymer as surface modifiers. By triggering the aggregation and de-aggregation of the magnetic particles through a magnetic field, they regulated the pH near the AMG molecules, achieving reversible control of the enzyme molecules [52]. When the immobilized AMG and Est are dispersed in an alkaline solution, the catalytic activity of AMG is inhibited. Under a low-frequency magnetic field (0.4 T), the nanoparticles aggregated. The immobilized Est reacted with ethyl acetate, providing an appropriate, slightly acidic environment (a local decrease in pH value) for AMG and activating AMG to catalyze the conversion of maltose to glucose. On the contrary, by replacing “Est” with “Ur”, it is possible to increase the pH near AMG under the influence of the magnetic field, leading to decreased activity of AMG. These experimental results indicate that an external magnetic field can activate the catalytic activity of the enzyme molecule by eliminating the unfavorable microenvironment near the substrate binding pocket or by counteracting the inhibitory effect of small molecules.

### 4.3. Enhanced Catalytic Activity of Magnetic Nanomaterials by Magneto-Electrical Effect

Since catalytic reactions often involve electron transfer, utilizing the magneto-electric effect to control the electronic structure of nanocatalysts has become a potential approach to enhancing their redox activity. The magnetostrictive-piezoelectric nanomaterial under an AMF can generate an electric potential on its surface through the magnetostrictive effect and the piezoelectric effect and can induce the separation of charge carriers [53]. For instance, Ge et al. prepared CoFe_2_O_4_-BiFeO_3_ magnetoelectric piezoelectric nanoparticles (CFO-BFO NPs) with a core–shell structure [23]. Under an AMF (B = 1.6 mT), the CFO core structure undergoes magnetostriction and generates microscopic stress. This causes a large number of vacancies and electrons to be generated on the surface of the BFO shell layer, triggering the production of a large number of •OH and superoxide anion radicals (•O^2−^) by H_2_O molecules and O_2_ molecules. The magnetic nanomaterials that integrate magnetostrictive and piezoelectric catalysis successfully induce efficient anticancer treatment. Zhang et al. synthesized FePt-FeC with a nano-heterogeneous structure and modified its surface with DSPE-PEG-TPP, enabling it to target to the mitochondria within cancer cells (Figure 6) [45]. Upon an AMF (f = 96 kHz, B ≤ 70 mT), a high density of induced charges is generated at the interface of the FePt-FeC heterostructure, which enhances the rates of NAD^+^ reduction reactions and Fenton-like reactions within the cells. Among them, the reduction of NAD^+^ can accelerate the aging of cancer cells at a rate of over 80%. This combined strategy has demonstrated that senescent cancer cells are also more susceptible to being killed by the generated NAD+ and •OH, resulting in excellent therapeutic effects both in vivo and in vitro. Although the current research on the mechanism of the magneto-electric effect is still in its early stages, magneto-electric-driven catalytic therapy has demonstrated its potential application in precise tumor treatment.

### 4.4. Enhanced Catalytic Activity of Magnetic Nanomaterials by Other External Energies

The catalytic activity of nanozymes with mixed redox states or shifting depends not only on the reaction conditions (such as temperature and pH) but also on the conversion frequency of their redox states according to the perspective of reaction kinetics. For example, Fe_3_O_4_ nanozymes have mixed redox states (Fe^2+^/Fe^3+^) in the H_2_O_2_ decomposing reaction. The Fe^2+^ active sites on the surface of the nanoparticle predominantly determine the catalytic activity. Accelerating the conversion frequency of (Fe^2+^/Fe^3+^) and promoting the reproduction of Fe^2+^ can enhance catalytic activity.

Employing external energy like X-rays to provide extra electrons to the redox state conversion cycling is a universal strategy to enhance the catalytic activity of nanozymes and for enzymes [46,54,55]. For instance, Zhang et al. synthesized a multicomponent SnS_2_@Fe_3_O_4_ nanoparticle and demonstrated the strategy for increasing catalytic activity by extra electron-promoted redox cycling under X-ray irradiation [47]. In detail, SnS_2_ as an electron donor can be excited by X-rays to transfer electrons to Fe_3_O_4_ owing to their matched electronic band (Figure 7), which promotes the transition from Fe^3+^ to Fe^2+^ and regeneration of Fe^2+^ on the surface of Fe_3_O_4_ nanoparticles. The accelerated redox cycling successfully resulted in Fe_3_O_4_ nanozymes maintaining higher catalytic activity to persistently produce more •OH for enhanced anticancer therapy.

Ultrasound can also penetrate deep into dermal tissues to enhance the catalytic activity of magnetic nanozymes at the tumor site. Developing a sonosensitive magnetic nanozyme would be a representative strategic approach that holds potential in clinical use [56]. Wang et al. rationally designed and synthesized a ZnFe_2_O_4_@Pt@PEG-GOx (ZFPG) nanozyme featuring five enzymatic activities, including POD, OXD, GOx, CAT, and glutathione oxidase (GSHOx) [43]. The nanozyme could effectively accumulate in the tumor under magnetic force induction. Subsequently, the sonosensitivity of the ZFPG nanozymes enhanced the depletion of the energy substrate (glutathione and glucose) and the production and utilization of H_2_O_2_, finally producing multiple ROS. In addition, rapid electron transfer from ZnF_2_O_4_ to Pt was observed under ultrasound irradiation. The creative method induces anti-tumor immunity to kill cancer cells and inhibit tumor metastasis (Figure 8). These findings, demonstrated through X-rays and ultrasonication, as compared to electromagnetic effects (as shown in Table 2), are also a good external energy for wireless and remote nanoparticles for noninvasive biomedical engineering.

## 5. Toxicity and Pharmacokinetics Effects of MNPs on Living Systems

Biological properties of MNPs, including pharmacokinetics and toxicity, have important influences on antitumor efficacy. This is because the nanoparticles after intravenous injection are recognized by the host immunological system and cleared from the body. For achieving an efficient accumulation of nanoparticles within the tumor, researchers should strike a balance between the clearance and the blood circulation of nanoparticles. Following antitumor applications, nanoparticles should be metabolized and eliminated from the body [57].

Knowing the characteristics of the used MNPs is crucial to improving their pharmacokinetics and minimizing their toxicity. The pharmacokinetics of nanoparticles are associated with their size, morphology, charge, and the nature of surface modifiers, offering different circulation times. For in vivo administration, the nanoparticles must be modified with hydrophilic molecules, such as dextran and polyethylene glycol, to ensure long-term stability and avoid aggregation. The size of nanoparticles is suggested to be controlled within the nanoscale range below 100 nm. The composition of nanoparticles should be generally designed to reduce their undesired toxic effects due to inevitable biodistribution. MNPs based on Fe, Mn, Co, and Mg exhibit different cytotoxicity in normal cells. Iron oxide nanoparticles are the most common type in antitumor applications, due to their low toxicity, biocompatibility, and long-term stability. The applications of iron oxide nanoparticles have been approved by the US Food and Drug Administration and the European Medicines Agency. Based on passive and active targeting effects, the iron oxide nanoparticles after administration can arrive and accumulate in the tumor region, showing reduced systemic toxicity. The nanoparticles in tissue can be further broken down into elemental iron by hundreds of iron metabolism-regulating proteins in cells. These iron ions are subsequently incorporated into natural iron element circulation in the body, thus reducing the toxicity of excessive accumulation of nanoparticles [5].

## 6. Perspectives and Conclusions

In this review, we aim to present the recent significant developments of various magnetic nanomaterials with enhanced enzymatic activities and their antitumor treatments over the past five years. Firstly, we introduced versatile MNPs and their fundamental magnetic properties and electromagnetic effects, including magneto-thermal, magneto-mechanical, and magneto-electric effects. Secondly, we summarized the MNPs’ enzymatic activities and advanced synthetic strategies, including nanozymes and immobilized enzymes. We highlighted the strategies and mechanisms of the enhanced enzymatic activities of magnetic nanomaterials through various electromagnetic effects in achieving efficient anticancer therapy. We also displayed some innovative magnetic nanomaterials with unique responsiveness to external energies like X-ray and ultrasound. Finally, we discussed the toxicity and pharmacokinetic effects of MNPs on living systems.

Despite the flourishing progress in developing MNPs with enhanced enzymatic activities, there are still important issues to be addressed. First, great efforts are needed to design novel magnetic nanomaterials with excellent magnetic properties. The structure–activity and magnetic field–activity relationships of magnetic nanomaterials should be systematically studied. This will enhance the controllability of catalytic reactions in vivo at high temporal precision. Second, it is important to explore the effective delivery and clearance methods of MNPs for their efficient and safe therapeutic applications. Third, it is necessary to investigate the mechanisms and verifying the validity of the external energy-enhanced catalytic processes and cascade catalytic reactions as well as the antitumor treatments of MNPs in large animal models. Finally, the development of bidirectional and effective regulatory strategies is a potential future direction for meeting the requirement of balancing catalytic reactions in living systems.

## Figures and Tables

**Figure 1 ijms-26-10890-f001:**
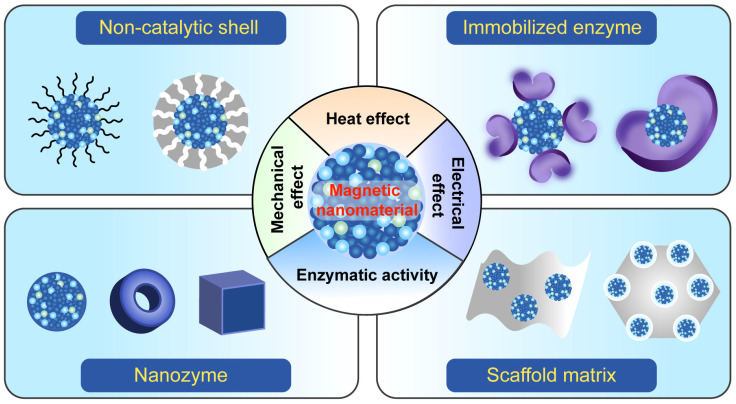
Advanced magnetic nanomaterials with enhanced enzymatic activities. The non-catalytic shell represents the surface modification of a nanomaterial and can include hydrophilic polymers, the mesoporous silica shell layer, and targeting molecules. We replaced the “natural enzyme” with the “immobilized enzyme,” which refers to the enzyme immobilized on magnetic nanomaterial carriers. The nanozyme includes various magnetic nanomaterials (such as nanoscale particles, rings, and cubes) that possess enzyme-like catalytic activity. The scaffold matrix refers to a magnetic catalytic reactor that employs matrix materials as a scaffold in loading nanozymes. Based on electromagnetic effects, including heat, mechanical, and electrical effects, magnetic nanomaterials under external magnetic fields exhibit enhanced enzymatic activities.

**Figure 2 ijms-26-10890-f002:**
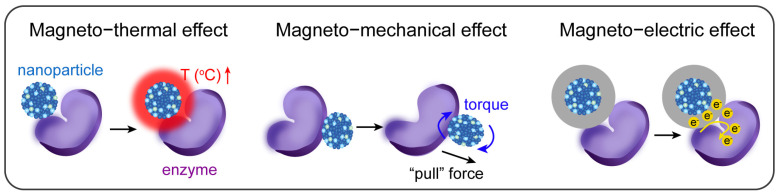
Diagrams of electromagnetic effects of magnetic nanoparticles (MNPs) for enhancing enzymatic activities. The magneto-thermal effect and magneto-electric effect are induced under alternating magnetic fields. The magneto-mechanical effect includes torque under uniform magnetic fields and “pull” force under magnetic field gradients.

**Figure 3 ijms-26-10890-f003:**
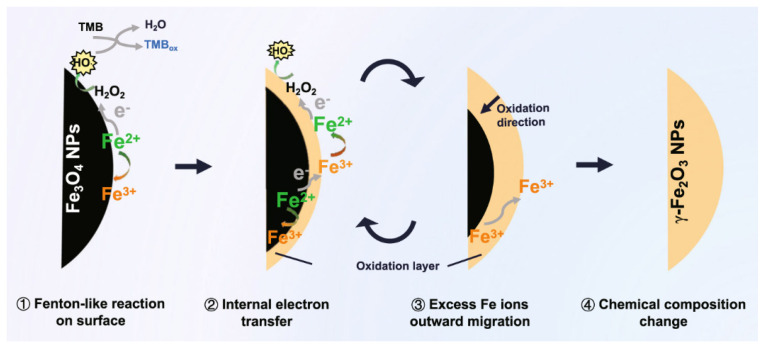
Illustration of the POD-like activity mechanism of the Fe_3_O_4_ nanozyme. Firstly, electrons from the surface Fe^2+^ are transferred to H_2_O_2,_ along with generating Fe^3+^. H_2_O_2_ is dissociated into •OH for oxidizing the 3,3′,5,5′-tetramethylbenzidine (TMB). Then, the internal Fe^2+^ contributes its electrons to the surface Fe^3+^, reproducing new Fe^2+^ on the surface of the nanoparticle for achieving sustained catalytic ability. Thirdly, excess Fe^3+^ generated during the oxidation of the internal Fe^2+^ migrates outward to the surface to maintain electroneutrality. Finally, the Fe_3_O_4_ nanoparticle is completely transformed into a γ-Fe_2_O_3_ nanoparticle from the surface to the internal part after repeated catalytic reactions. Reprinted with permission from Ref. [33]. Copyright (2022) Springer Nature.

**Figure 4 ijms-26-10890-f004:**
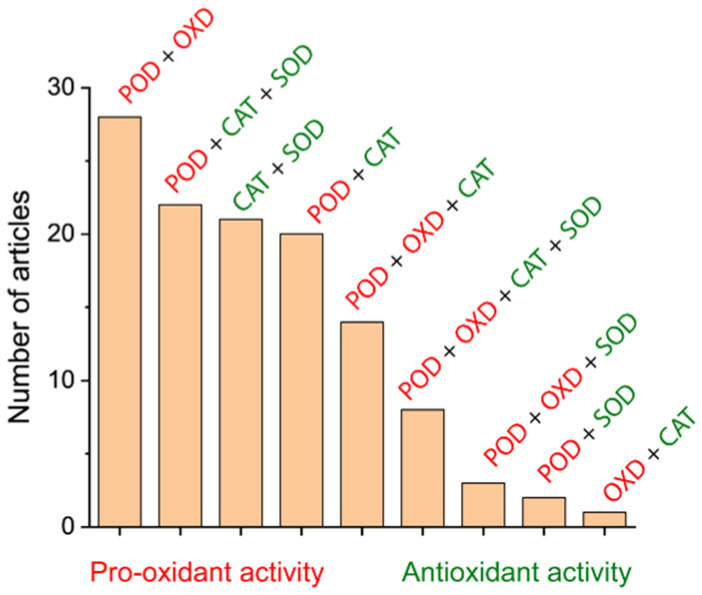
Nanozymes exhibit simultaneous pro-oxidant and antioxidant activities. Reprinted with permission from Ref. [32]. Copyright (2025) John Wiley and Sons.

**Figure 5 ijms-26-10890-f005:**
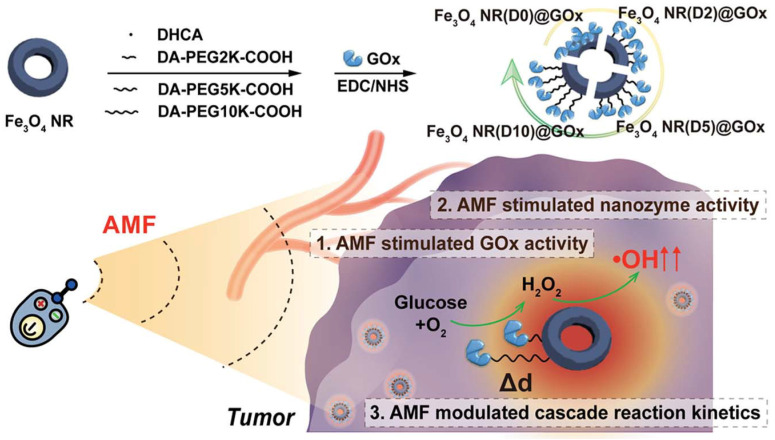
Schematic diagram of a nanozyme-enzyme multicomponent cascade catalyst, namely the Fe_3_O_4_ NR-glucose oxidase (Fe_3_O_4_ NRs@GOx) system. Dopamine-PEG-COOH with different molecular weights (including 0 (D0), 2K (D2), 5K (D5), and 10K (D10)) was functionalized on the surface of FVIOs for immobilizing GOx. The cascade catalytic activity for hydroxyl radical generation was regulated by the magneto-thermal effect for the doplasmic reticulum. Reconstructed with permission from Ref. [42]. Copyright (2021) American Chemical Society.

**Figure 6 ijms-26-10890-f006:**
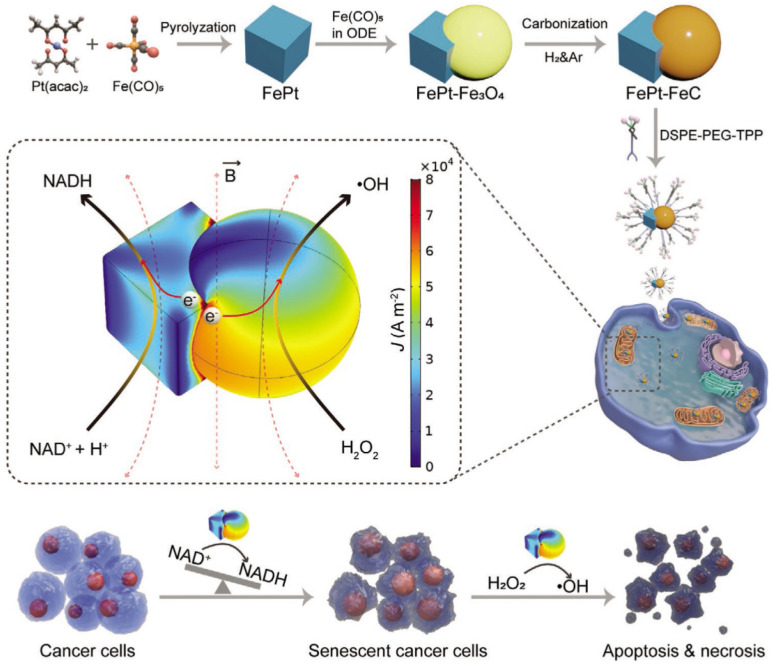
Illustration of the synthetic method and the magneto-electro catalytic therapeutic strategy of FePt-FeC heterostructures. FePt-FeC heterostructures generate a high-density induced current at the interface under an AMF. This produces a higher efficiency of NAD^+^ reduction and Fenton-like reaction, which results in the senescence and apoptosis of cancer cells. Reprinted with permission from Ref. [45]. Copyright (2021) John Wiley and Sons.

**Figure 7 ijms-26-10890-f007:**
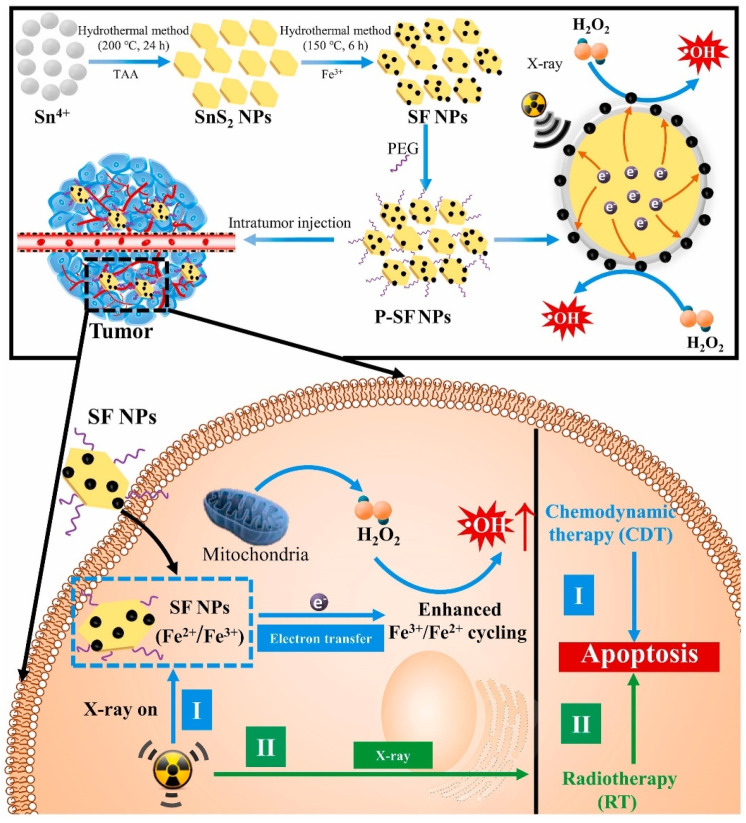
The mechanism of multicomponent SnS_2_@Fe_3_O_4_ nanozymes with POD-like activity showing ROS-enhanced ability under X-ray exposure for anticancer therapy. Reconstructed with permission from Ref. [47]. Copyright (2021) Elsevier.

**Figure 8 ijms-26-10890-f008:**
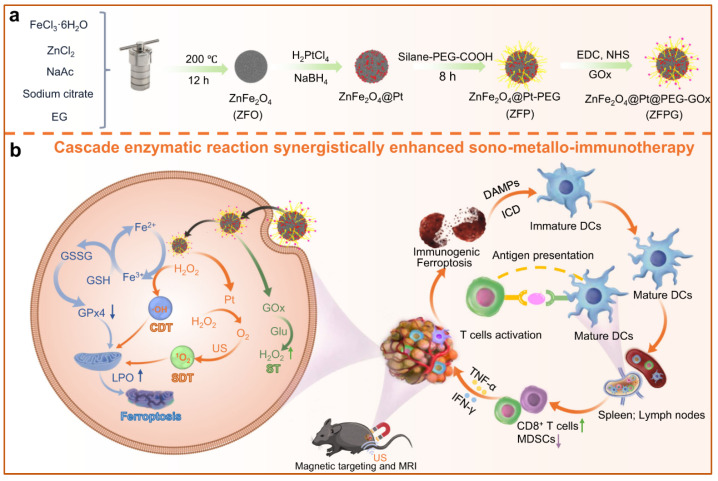
Schematic illustration of (**a**) the synthetic method of ZFPG nanozymes and (**b**) the anti-tumor mechanism based on enhanced cascade catalytic activities under ultrasound irradiation and sono-metallo-immunotherapy for prostate cancer treatment. Reconstructed with permission from Ref. [43]. Copyright (2025) Springer Nature.

**Table 1 ijms-26-10890-t001:** Examples of magnetic nanomaterials with enzymatic activities in antitumor applications.

Nanomaterial	Enzymatic Activity	Mechanism	Application
Fe_3_O_4_-GOx in silica nanoparticles [37]	Showed rapid glucose consumption and abundant •OH production	Cascade catalytic reactions and reduced pH	Resulting in both “starvation” and oxidative antitumor effects in mice bearing 4T1 breast cancer
Magnetic covalent organic framework confined Os nanoclusterzyme Fe_3_O_4_@COF@Os [27]	Displayed superior POD-specific activity	Regulated by deoxidizers and the functional groups	Analyze serum cancer biomarker prostate-specific antigen, with a detection limit of 3.83 pg mL^-1^
Porphyrin-Zr-MOF-zinc ferrite-based magnetic nanozyme [28]	Showed multienzyme-like cascade activity	Magneto-electrical effect	MOF-based magnetic nanozyme exhibits significantly enhanced ferroptosis in tumor-bearing mice
Fe_3_O_4_@ZIF-67@Pt nanozyme [29]	Showed significantly enhanced peroxidase-like activity	Synergistic effect between ZIF-67 and Pt	Achieved a highly accurate detection of PSA in human serum samples and distinguished prostate cancer patients from healthy individuals
Hollow Fe_3_O_4_ mesocrystals [31]	Exhibited increased peroxidase-like activity	High ratio of Fe^2+^/Fe^3+^, many oxygen defects, and magneto-thermal effect	Achieved a self-augmented synergistic effect between hyperthermia and catalytic therapy of 4T1-tumor-bearing nude mice
Magnetic Fe_3_O_4_ nanocomposites [34]	Showed enhanced catalase-like activity	Magneto-thermal effect	Achieved tumor angiogenesis inhibition in BALB/c nude mice bearing fLuc-LM3 orthotopic HCC
Yolk–shell Fe_2_C@Fe_3_O_4_-PEG nanozymes [38]	Showed high magnetothermal-enhanced peroxidase-like activities	Magneto-thermal effect	Exhibited superior synergistic antibacterial efficacy in in vitro and in vivo experiments
Fe_3_O_4_ nanoparticles [39]	First reported magneto-thermal regulation on nanozyme activity	Local magneto-thermal effect	-
Fe_3_O_4_ nanoring and graphene oxide hybrid nanoparticles [40]	Induced a significantly amplified ROS level	Magneto-thermal effect	Attained ROS-related immune response and impressive systemic therapeutic efficacy of 4T1-tumor-bearing mice
Ir@MnFe_2_O_4_ nanozyme [41]	Increased the rate of conversion of both Fe(III) to Fe(II) and H_2_O_2_ to •OH	Magneto-thermal effect	Mediated cellular redox homeostasis disruption and led to efficient treatment in mice bearing xenografted HeLa tumors
GOx-Fe_3_O_4_ nanoring [42]	Exhibited a superior kinetic match between GOx and Fe_3_O_4_ nanozymes and over a 400-fold higher cascade activity	Magneto-thermal effect, cascade catalytic reaction	Significantly improved tumor inhibition in 4T1 tumor-bearing mice.
ZnFe_2_O_4_@Pt-GOx core-shell nanoparticles [43]	Showed five-enzyme activities	Ultrasound activation, cascade catalytic reaction	Facilitated sono-metallo-immunotherapy for prostate cancer treatment in male mice.
Magnetic nanoparticle-HRP-silica nanohybrids [44]	Increased HRP-catalyzed conversion rate from indole-3-acetic acid into peroxylated radicals	Magneto-thermal effect, cascade catalytic reaction	Showed higher reductions in the tumor volume growth in mice using a human pancreatic cancer cell line
FePt-FeC heterostructures [45]	Enhanced catalytic hydrogenation reaction of FePt-FeC heterostructures	Magneto-electrical effect	Killed senescent cancer cells efficiently with catalytic therapy
Carbon-coated NiFe_2_O_4_ nanocatalysts [46]	Increased Fenton reaction activity	Localized electron distribution at octahedral Fe active sites	Mediated efficient photothermal and catalytic therapy in 4T1 tumor-bearing mice
SnS_2_-Fe_3_O_4_ nanocomposites [47]	Enhanced catalytic reaction of overexpressed H_2_O_2_ to ROS	X-ray-mediated electron transfer.	Demonstrated synergistic effects of radiotherapy and catalytic therapy in HeLa tumor-bearing nude mice

**Table 2 ijms-26-10890-t002:** The advantages and limitations of the electromagnetic effects and other energies used for enhancing the enzymatic activities of magnetic nanomaterials.

Method	Advantage	Limitation
Magneto-thermal effect	Catalytic therapy combined with magnetic hyperthermia	Enzyme inactivation is possible
		Overheating damage is possible
Magneto-mechanical effect	Low field strength is possible Can be used for drug delivery	Very large gradients are needed
Magneto-electrical effect	Effective for enhancing the redox reaction	Working distance is usually short
		Force/torque may be insufficient
X-rays	No limit on penetration depth	Inevitable radiation damage
	Long working distance	
Ultrasound	Acoustic wave induced pressure and heat	Working distance is short
		Hard to penetrate bone

## Data Availability

No new data were created or analyzed in this study. Data sharing is not applicable to this article.

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
