# Peer review of "Recent Progress of Magnetic Nanomaterials with Enhanced Enzymatic Activities in Antitumor Therapy"

_ijms, 2025, doi:10.3390/ijms262210890_

Round 1

Reviewer 1 Report

Comments and Suggestions for Authors

The article is interesting and addresses an important issue — the enzymatic activity of magnetic nanoparticles, particularly in the context of anticancer therapy. However, revisions are necessary.

Major revisions:

  • Scheme 1 – it is far too uninformative. The terms “Non-catalytic shell,” “Natural enzyme,” “Nanoenzyme,” and “Scaffold matrix” should be clearly explained in the scheme description.
  • Figure 1 – similarly, the caption merely repeats the labels shown in the image; it should provide a more detailed explanation. What does TMB stand for?
  • Section 3.2. Magnetic nanomaterial-immobilized enzymes – this part is far too brief and lacks sufficient detail. Please include specific examples, mechanisms, and applications.
  • Most importantly, the manuscript lacks a final section summarizing the mechanisms of nanozyme activity and their biological effects on cells, as well as examples of applications in different types of cancers. Are any of these approaches already in clinical use?
  • It is also recommended to include tables to systematically organize the information presented in the text.

Minor revisions:

  • Why is the first illustration titled Scheme 1 while the next is Figure 1? All illustrations should be consistently labeled as Figures from the beginning.
  • Lines 94–95: “Therefore, the magnetic field-induced electromagnetic effects of nanoparticles are extremely ingenious approaches to achieve ideal noninvasive cancer therapy.” — The term “ideal” seems overly strong and should be toned down.
  • Lines 100–103: The same information is repeated in two consecutive sentences and should be condensed.
  • Lines 160–163 vs 166–169: These sections repeat the same content and should be revised for conciseness.

Reviewer 2 Report

Comments and Suggestions for Authors

The manuscript ID: ijms-3966158: “Recent progress of magnetic nanomaterials with enhanced enzymatic activities in antitumor therapy” by Zhang, Y.; et al. is a Review work where the authors outlined the most relevant advances in the field of magnetic nanomaterials and how their electromagnetic properties can ameliorate the catalytic activities against cancer diseases.

Nevertheless, the below described points need to be fully addressed to be consider for further publication:

1 Introduction. “Magnetic nanomaterials with enzymatic activities (…) in cancer catalytic therapy” (lines 32-34). Could the authors provide quantitative data insights according to the worldwide global burdens of cancer and the linked disability-adjusted life years (DALYs)? This will significantly aid the potential readers to better understand the significance of this Review work.

  1. “Various magnetic nanoparticles (MNPs) and their electromagnetic effects” (line 76-149). Here, the authors show the electromagnetic effects of certain nanoparticles and how some clinical therapies exploit them for drug delivery purposes. Have been reported accumulation effects after the MNP administration? Some discussion should be furnished in this regard.

  1. “Design and synthesis of magnetic nanomaterials with enzymatic activities” (lines 150-334). What are the reaction yields for all the mentioned magnetic nanomaterial synthetic routes? Is it possible to shape the dimensions and morphology of the magnetic nanomaterials during their sinthesis and could this aspect strongly impact on their electromagnetic properties?

  1. “Enhance enzymatic activities of magnetic nanomaterials for efficient antitumor therapy (lines 335-541). Here a table highlighting the main advantages and limitations related to the magneto-thermal, magneto-mechanical, magneto-electrical effects and other energies should be added.

  1. Finally, the authors should briefly discuss about nanoscale imaging tools [1] and magnetic resonance imaging [2] can assist in the cancer diagnosis and detection and how combined with magnetic nanomaterials design the next-generation of prognosis and anticancer clinical therapies.

[1] https://doi.org/10.1002/smsc.202500351

[2] https://doi.org/10.1007/s00330-025-11976-5

6 “Perspectives and Conclusions” (lines 542-570). This section perfectly remarks the most relevant outcomes found by the authors in this field and the promising future prospectives. It may be also opportune to highlight the potential future action lines to pursue the topic covered in this work.

Author Response

Comments 1: Introduction. “Magnetic nanomaterials with enzymatic activities (…) in cancer catalytic therapy” (lines 32-34). Could the authors provide quantitative data insights according to the worldwide global burdens of cancer and the linked disability-adjusted life years (DALYs)? This will significantly aid the potential readers to better understand the significance of this Review work.

Response 1: Thanks a lot for the suggestions. We have added the quantitative data in the revised manuscript.

The global cancer burden continues to rise and growth. There were approximately 18.5 million new cancer diagnoses and 10.4 million cancer deaths in 2023, resulting in a total of 271 million disability-adjusted life-years (DALYs). The number of cancer diagnoses and deaths was projected to rise substantially from 2024 to 2050, by 60.7% and 74.5%, respectively. This will result in an estimated 30.5 million new cancer diagnoses and 18.6 million deaths by 2050 [1,2].

Comments 2: “Various magnetic nanoparticles (MNPs) and their electromagnetic effects” (line 76-149). Here, the authors show the electromagnetic effects of certain nanoparticles and how some clinical therapies exploit them for drug delivery purposes. Have been reported accumulation effects after the MNP administration? Some discussion should be furnished in this regard.

Response 2: Thank the reviewer for your kind comment.

Two clinical studies about magnetic drug delivery is shown in clinicaltrial.gov database [3]. For example, the superparamagnetic iron oxide nanoparticles are injected for neoadjuvant chemotherapy of Osteosarcoma (NCT04316091). The magnetic beads loaded with doxorubicin are delivered for magnetic-guided chemotherapy of colorectal neoplasm (NCT00041808). However, there are still no available results about the accumulation effects of these magnetic nanoparticles after administration.

Comments 3: “Design and synthesis of magnetic nanomaterials with enzymatic activities” (lines 150-334). What are the reaction yields for all the mentioned magnetic nanomaterial synthetic routes? Is it possible to shape the dimensions and morphology of the magnetic nanomaterials during their sinthesis and could this aspect strongly impact on their electromagnetic properties?

Response 4: Thanks a lot for your suggestion. The reaction yield values of all the mentioned synthetic routes were not reported or varied in different studies. All we know is the ranking of the yields of various synthetic methods. The reaction yields of co-precipitation, thermal decomposition, and hydrothermal methods are medium, high, and high, respectively [4].

The dimensions and morphology of magnetic nanomaterials have a great impact on their electromagnetic properties. Optimizing the dimensions and morphology of nanomaterials is crucial to improving their electromagnetic properties and biomedical applications. Chemical synthesis methods such as thermal decomposition and hydrothermal reaction are commonly used [5,6]. The size of nanoparticles is closely associated with the reaction time and the boiling point of the organic solvent during thermal decomposition [7]. Controlling reaction time and temperature can restrict the overgrowth of nanoparticles. Using organic ligands (such as oleylamine) can shape their morphology. We hava added discussions in the revised manuscript.

Comments 4: “Enhance enzymatic activities of magnetic nanomaterials for efficient antitumor therapy (lines 335-541). Here a table highlighting the main advantages and limitations related to the magneto-thermal, magneto-mechanical, magneto-electrical effects and other energies should be added.

Response 4: Thanks a lot for your suggestion. We have added a table (as shown in Table 1) highlighting the main advantages and limitations related to the magneto-thermal, magneto-mechanical, magneto-electrical effects, and other energies in the revised manuscript.

Table 1. The advantages and limitations of the electromagnetic effects and other energies used for enhancing the enzymatic activities of magnetic nanomaterials.

Methods

Advantages

Limitations

Magneto-thermal effect

Heating rate is easy to control. Catalytic therapy combined with magnetic hyperthermia.

Enzyme inactivation is possible.

Overheating damage is possible.

Magneto-mechanical effect

Low field strength is possible.

Can be used for drug delivery.

Very large gradients are needed.

Magneto-electrical effect

Effective for enhancing the redox catalytic activity.

Working distance is usually short.

Force/torque may be insufficient

X-rays

No limit on penetration depth.

Long working distance.

Inevitable radiation damage.

Ultrasounds

Acoustic wave induced pressure and heat.

Working distance is short.

Hard to penetrate bone.

Comments 5: Finally, the authors should briefly discuss about nanoscale imaging tools [1] and magnetic resonance imaging [2] can assist in the cancer diagnosis and detection and how combined with magnetic nanomaterials design the next-generation of prognosis and anticancer clinical therapies. [1] https://doi.org/10.1002/smsc.202500351; [2] https://doi.org/10.1007/s00330-025-11976-5.

Response 5: Thank you for your suggestions. We have added the discussions about nanoscale imaging tools and resonance imaging that assisted the cancer diagnosis and detection. We also provide a brief discussion about designing magnetic nanomaterials for the next generation of prognosis and antitumor clinical therapies. We have added these references in the revised manuscript.

Efficient antitumor therapy underscores the need for imaging the sufficient accumulation of magnetic nanoparticles within tumors. Magnetic resonance imaging (MRI) enables high-contrast imaging of tumor tissue. The whole-body MRI is a promising noninvasive modality for cancer diagnosis [8,9]. However, WB-MRI still possess modest detection rates. Magnetic nanoparticles, one of the most used nanoscale imaging tools, can serve as MRI contrast agents for improving cancer detection rates. According to biomarkers of cancer, the magnetic nanoparticles can be modified with various antibodies or molecules and then actively targeted to tumors following administration. MRI can be used to confirm the maximum accumulation of the nanoparticles in the tumor site. Based that, the improved efficacy of the nanoparticle-mediated antitumor catalytic therapy can be achieved.

Comments 6: “Perspectives and Conclusions” (lines 542-570). This section perfectly remarks the most relevant outcomes found by the authors in this field and the promising future prospectives. It may be also opportune to highlight the potential future action lines to pursue the topic covered in this work.

Response 6: Thank you for your suggestions. We have added discussions about the potential future action in the revised manuscript.

Despite the flourishing progress in developing magnetic nanoparticles with enhanced enzymatic activities, there are still important issues to be addressed. First, great efforts are expected to design novel magnetic nanomaterials with excellent magnetic properties. This will enhance the controllability of catalytic reactions in vivo at high temporal precision. Second, exploring effective delivery and clearance methods of magnetic nanoparticles for their efficient and safe therapeutic applications. Third, investigating the mechanism and verifying the validity of external energy-enhanced catalytic processes and cascade catalytic reactions, as well as antitumor treatments of magnetic nanoparticles in large animal models. Finally, developing bidirectional and effective regulatory strategies would be a potential future direction for meeting the requirement of balancing a certain catalytic reaction in living systems.

4. Response to Comments on the Quality of English Language

Point 1: No.

Response 1:

5. Additional clarifications

No.

References

[1]         Luo, Q.; Smith, D. Global cancer burden: progress, projections, and challenges. Lancet, 2025, 406(10512):1536-1537.

[2]         GBD 2023 Cancer Collaborators. The global, regional, and national burden of cancer, 1990-2023, with forecasts to 2050: a systematic analysis for the Global Burden of Disease Study 2023. Lancet, 2025, 406(10512):1565-1586.

[3]         Liang, C.; Zhang, X.; Cheng, Z.; Yang, M.; Huang, W.; Dong, X. Magnetic iron oxide nanomaterials: a key player in cancer nanomedicine. View, 2020, 1(3): 20200046.

[4]         Zhao, S.; Yu, X.; Qian, Y.; Chen, W.; Shen, J. Magnetic nanoparticles: synthesis, anisotropy, and applications. Chemical Reviews, 2023, 123(7): 3904-3943.

[5]         Rezaei, B.; Yari, P.; Sanders, S.; Wang, H.; Chugh, V.; Liang, S.; Mostufa, S.; Xu, K.; Wang, J.; Gómez-Pastora, J.; Wu, K. Magnetic nanoparticles: a review on synthesis, characterization, functionalization, and biomedical applications. Small, 2024, 20(5): 2304848.

[6]         Wang, S.; Xu, J.; Li, W.; et al. Magnetic nanostructures: Rational design and fabrication strategies toward diverse applications. Chemical reviews, 2022, 122(6): 5411-5475.

[7]         Ma, Z.; Mohapatra, J.; Wei, K.; Liu, J.; Sun, S. Magnetic nanoparticles: synthesis, anisotropy, and applications. Chemical Reviews, 2023, 123(7): 3904-3943.

[8]         Fonseca, J.; Trennepohl, T.; Pinheiro, L.; Forte, G.; Campello, C.; Altmayer, S.; Andrade, R.; Hochhegger, B. Whole-body MRI for opportunistic cancer detection in asymptomatic individuals: a systematic review and meta-analysis. European Radiology, 2025: 1-11.

[9]         Marcuello, C.; Lim, K.; Nisini, G.; Pokrovsky, V.; Conde, J.; Ruggeri F. Nanoscale analysis beyond imaging by atomic force microscopy: molecular perspectives on oncology and neurodegeneration. Small Science, 2025, 2500351.

Reviewer 3 Report

Comments and Suggestions for Authors

The work represents a high-quality systematic review of the state-of-the-art and future perspectives for magnetic nanomaterials with regulatory catalytic functions in antitumor therapy. The content is of interest to a broad range of specialists in nanomaterials, biocatalysis, and medicinal chemistry, and can also serve as a starting point for further research. The relevance of the chosen topic is beyond doubt, and the research area itself has been developing extremely dynamically in recent years.

There are several remarks:

  1. The review lacks a dedicated section on the biological properties of nanoparticles—specifically, toxicity, pharmacokinetics, and long-term effects on living systems.

  2. To improve the clarity and illustrative value of the article, it would be advisable to add diagrams of magnetic effects (magnetothermal, magnetoelectric, magnetomechanical, and combined modalities).

Round 2

Reviewer 1 Report

Comments and Suggestions for Authors

Dear Authors, I’m glad that you have addressed my comments and improved your manuscript. Wishing you success in your research work.

Reviewer 2 Report

Comments and Suggestions for Authors

The authors did a great effort and the manuscript quality was greatly improved. Based on the significance of this Review work, I warmly endorse it for publication in its current form